# Validity and Cost-Effectiveness of Pediatric Home Respiratory Polygraphy for the Diagnosis of Obstructive Sleep Apnea in Children: Rationale, Study Design, and Methodology

**DOI:** 10.3390/mps4010009

**Published:** 2021-01-19

**Authors:** Esther Oceja, Paula Rodríguez, María José Jurado, Maria Luz Alonso, Genoveva del Río, María Ángeles Villar, Olga Mediano, Marian Martínez, Santiago Juarros, Milagros Merino, Jaime Corral, Carmen Luna, Leila Kheirandish-Gozal, David Gozal, Joaquín Durán-Cantolla

**Affiliations:** 1Domiciliary Hospitalization, Sleep Unit, OSI Araba University Hospital, 01004 Vitoria, Spain; esther.ocejabarrutieta@osakidetza.eus; 2Research Service and Bioaraba Research Institute, OSI Araba University Hospital, UPV/EHU, 01004 Vitoria, Spain; joaquin.durancantolla@gmail.com; 3Sleep Unit, Hospital Universitario Valle de Hebrón, 08035 Barcelona, Spain; mjjuradoluque@gmail.com; 4Sleep Unit, Complejo Hospitalario de Burgos, 09006 Burgos, Spain; 5Sleep Unit, Fundación Jiménez Díaz, 28040 Madrid, Spain; GRCamacho@fjd.es; 6Sleep Unit, Hospital Universitario de Basurto, 48013 Bilbao, Spain; mariaangeles.villaralvarez@osakidetza.eus; 7Sleep Unit, Hospital de Guadalajara, 19002 Guadalajara, Spain; olgamediano@hotmail.com; 8Sleep Unit, Hospital Universitario Marqués de Valdecilla, 39008 Santander, Spain; marian15m@hotmail.com; 9Sleep Unit, Hospital Universitario de Valladolid, 47012 Valladolid, Spain; santiagojuarros@separ.es; 10Sleep Unit, Hospital Universitario La Paz, 28046 Madrid, Spain; mmerinoa@salud.madrid.org; 11Sleep Unit, Complejo Hospitalario de Cáceres, 100003 Cáceres, Spain; jcorral@separ.es; 12Sleep Unit, Hospital Universitario 12 de Octubre, 280035 Madrid, Spain; lunalavin1@yahoo.es; 13Department of Child Health and Child Health Research Institute, School of Medicine, University of Missouri, Columbia, MO 65201, USA; lkheurandish-Gozal@peds.bsd.uchicago.edu (L.K.-G.); dgozal@peds.bsd.uchicago.edu (D.G.)

**Keywords:** obstructive sleep apnea (OSA), children, diagnosis, respiratory polygraphy, cost-effectiveness analysis

## Abstract

Obstructive sleep apnea (OSA) in children is a prevalent, albeit largely undiagnosed disease associated with a large spectrum of morbidities. Overnight in-lab polysomnography remains the gold standard diagnostic approach, but is time-consuming, inconvenient, and expensive, and not readily available in many places. Simplified Home Respiratory Polygraphy (HRP) approaches have been proposed to reduce costs and facilitate the diagnostic process. However, evidence supporting the validity of HRP is still scarce, hampering its implementation in routine clinical use. The objectives were: Primary; to establish the diagnostic and therapeutic decision validity of a simplified HRP approach compared to PSG among children at risk of OSA. Secondary: (a) Analyze the cost-effectiveness of the HRP versus in-lab PSG in evaluation and treatment of pediatric OSA; (b) Evaluate the impact of therapeutic interventions based on HRP versus PSG findings six months after treatment using sleep and health parameters and quality of life instruments; (c) Discovery and validity of the urine biomarkers to establish the diagnosis of OSA and changes after treatment.

## 1. Introduction

Obstructive sleep apnea (OSA) is a common disorder of children characterized by habitual snoring, prolonged episodes of increased upper airway resistance, and respiratory effort with partial or complete upper airway obstruction during sleep. OSA is accompanied by different degrees of intermittent hypoxemia and hypercapnia and can lead to sleep fragmentation [1,2,3,4]. OSA affects 1–4% of children with a peak incidence around 2–8 years [1,2,4] and most of children with OSA are under-recognized [5]. The severity of OSA is measured by the apnea-hypopnea index (AHI), whereby most centers consider a AHI ≥ 3 events/h as clinically significant, and will routinely initiate treatment [1,2]. In the last several decades since its initial description, OSA has been associated with a large spectrum of neurocognitive, behavioral, cardiovascular, and metabolic adverse consequences, and it has been assumed that timely recognition of OSA and its resolution through adequate treatment would eventually reverse or prevent these consequences [5,6,7]. The most effective treatment of OSA in children consists in surgical adenotonsillectomy (T&A), although several alternative treatments have been developed in recent years [6,7].

The recommended diagnostic approach of OSA in children consists in conventional overnight polysomnography (PSG) in the sleep laboratory [8]. However, PSG is expensive, time-consuming, and inconvenient for patients and parents. In addition, children and caretakers are required to spend the night in the laboratory, an unfamiliar environment that can generate anxiety and alter sleep architecture. On the other hand, PSG is not readily available in many places around the world, and the overall number of professionals with expertise in sleep medicine further restricts the accessibility, leading to substantial delays and waiting times [9]. 

In an effort to improve upon the delays imposed by the aforementioned circumstances, simplified approaches have been explored. However, medical history and physical examination or commonly used questionnaires [6,10,11,12] are insufficiently accurate in the process of reaching a diagnosis. Similarly, video and audio recordings have proven as invalid [13,14]. Overnight oximetry recordings at home are comfortable, simple, and cheap, but diagnostic accuracy becomes particularly problematic when the severity of OSA is mild, even if recent artificial intelligence methodologies have markedly improved the reliability of this approach [15,16]. Nevertheless, in selected cases of suspected severe OSA, and when unavoidable delays due to PSG unavailability are likely, simple overnight oximetry can be successfully used [17,18]. 

Home respiratory polygraphy (HRP) has been proposed to reduce costs and facilitate the diagnostic process. This method has been widely validated and implemented in the adult population, and has become a realistic alternative for the diagnosis of suspected OSA in this age group [19]. However, the validation of HRP in children with suspected OSA has been less definitive, and has yet to allow for its widespread implementation in the context of the diagnosis of childhood OSA [20,21,22,23,24,25,26]. Indeed, a recent consensus from the American Academy of Sleep Medicine provided an unsupportive statement to the widespread use of HRP in children [27]. Several studies have been published on the validity of HRP in children with discordant results, and the vast majority of the studies was based on relatively small patient cohorts [28,29,30,31,32,33,34,35,36]. A systematic review concluded that although there is limited evidence concerning diagnostic alternatives to PSG for identifying OSA in children, polygraphy rather than PSG may be a valid test, a conclusion that will require more extensive confirmation in subsequent studies [37]. In addition, the role of HRP as a follow-up approach after treatment has not been thoroughly investigated [38]. 

Apnea-Link Air (Resmed^®^Spain) is a simplified polygraphic recording device designed for domiciliary use. It consists of a nasal canula with a pressure transducer, with thoracic and abdominal effort being measured by plethysmography, and oxyhemoglobin saturation by continuous pulse oximetry. The proprietary software of the device allows for both automatic and manual analysis of the physiologic signals. Its use has been validated in adults [39,40,41], but its utility in children has yet to be explored.

In the quest to develop simplified approaches to the diagnosis of OSA in children and potentially identify OSA-associated morbidities, exploration of biomarkers was pioneered about a decade ago, with overall encouraging findings [42,43,44,45,46,47,48,49,50,51,52]. Indeed, a large number of candidate biomarkers consisting of proteins and microRNAs have emerged, and have provided important insights into potential mechanisms underlying OSA-associated morbidities while also displaying favorable diagnostic profiles. 

Based on aforementioned considerations, the present study protocol was designed to address several critically important issues aimed at facilitating access to symptomatic children at risk for OSA, enabling their diagnosis at home, identifying associations with pragmatically applicable cognitive testing, identify exosomal miRNA markers in their urine, and ultimately allow for evaluation of their treatment outcomes. 

The main objective of the study was to assess the diagnostic validity of a HRP, specifically the Apnea-Link Air system, in children with clinically suspected OSA, and compare the findings with concurrent PSG results. 

The secondary outcomes of the proposed study were to:Analyze the cost-effectiveness of HRP compared with PSG.To determine the validity of diagnostic and therapeutic decisions using the results of HRP findings based on either manual scoring or automatic analysis, compared to decisions using PSG results.To identify and analyze the validity urine biomarkers in the diagnosis and in identification of cognitive morbidity among children with OSA children and their changes after OSA treatment.

## 2. Materials and Methods

Design: Randomized, prospective, multicenter, double blind and crossover trial of a targeted clinical referral cohort of 320 symptomatic children (ages between 2 and 14 years).

Measurements: Children clinically suspected for OSA will be evaluated by clinical history and anthropometric measures, along with validated clinical questionnaires about OSA symptoms, quality of life, and co-morbidities. All children will undergo HRP at home and PSG in the sleep laboratory at the beginning of the study and repeated evaluation six months after treatment. State-of-the-art exosome urine analysis for miRNA biomarkers will be conducted. Cost-effectiveness of HRP vs. PSG will be analyzed.

Interest of trial: Largest unbiased multicenter study for simplified HRP validation in children which should also provide important insights into outcomes and cost-effectiveness along with discovery of OSA-relevant urinary biomarkers in children. 

### 2.1. Study Design

The DINISAS project is a multicenter, randomized, cross-over study aimed at analyzing the validity of a HRP in children with suspected OSA compared to overnight PSG performed in the sleep laboratory. This analysis will be performed using blinded outcome assessments, and the project was initiated in 2014. 

### 2.2. Protocol

#### 2.2.1. Subjects

Children from 2 to 14 years of age of both sexes consecutively referred for clinical suspicion of OSA will be selected if they meet the inclusion and exclusion criteria listed in Table 1. Clinical suspicion of OSA is defined as: snoring children with respiratory pauses and/or observed apneas and/or ventilatory effort during sleep as observed by parents, and for whom sleep testing was requested by their primary care physician.

#### 2.2.2. Randomization and Follow Up

All selected children will undergo randomization as follows: Group 1 will be children who whose initial diagnostic test will be home polygraphy (HPR) followed by in-lab PSG; and Group 2 will include children whose initial testing will be in-lab PSG, subsequently followed by HRP. Time interval between the two tests, HRP and PSG, will always be less than one week. Therapeutic decisions (surgical adenotonsillectomy (T&A), continuous positive airway pressure (CPAP), orthodontics, or medical treatment) will be based on the apnea-hypopnea index derived from the HRP, with all PSG results being invisible to both investigators and participating subjects. An independent Safety Monitoring Board with extensive expertise in pediatric OSA was appointed and reviewed both HRP and PSG and if decisions regarding the specific subject based on HRP were inappropriate, the correct intervention was provided (i.e., surgery vs. no surgery).

An overnight in-lab PSG will be performed in all randomized OSA children six months after treatment or simply as a follow-up if no treatment was provided.

Questionnaires and urine tests were conducted in the morning following PSG testing. 

Flow chart of randomization procedures and follow-up of children are shown in Figure 1.

#### 2.2.3. Variables 

All collected variables for the study are shown in Table 2. 

### 2.3. Anthropometric and Others Clinical Variables

In the pre-randomization stage, collection of sociodemographic data of each child including age, gender, any comorbidity, and findings from physical examination, including the degree of tonsil hypertrophy, Mallampati score, and maxillomandibular profile will be recorded. Anthropometric data will be obtained after each of the two PSG tests. We will measure in each subject, weight, height, body mass index (and percentiles), neck and waist circumference, and arterial blood pressure (ABP). ABP will be measured 3 times after 5 min of resting in the supine position using standardized and validated automatic sphygmomanometer (Spacelab type) and using the appropriate arm cuff size for the child.

### 2.4. Sleep Studies

#### 2.4.1. Polysomnography in Sleep Laboratory (PSG)

An overnight PSG will be performed at baseline and at six months based on the American Academy of Sleep Medicine guidelines [53], the recent European Respiratory Society recommendations [54], and the Spanish Sleep Apnea Recommendations [55]. EEG recordings will include six electroencephalographic electrodes: two frontal (F3–F4), two central (C3–C4) and two occipital (O1–O2) locations, referred to contralateral mastoids (A1–A2) and adopting the 10–20 rules of international EEG system. We will include one ground electrode and another reference electrode (Cz). The electromyogram will be obtained using two chin electrodes and the electrooculogram (EOG) will be recorded employing two different electrodes placed above the left and right outer eye cantus. The different sleep stages: N1, N2, N3 (NREM sleep), and REM will be scored; arousals, oxygen saturation (SaO_2_) and apneas and hypopneas were identified (Table 3) using standardized guidelines [56,57,58,59]. The PSG will be considered as valid if total recording time is ≥300 min and includes at least 180 min of actual sleep. All the technicians scoring PSG will be blinded to the results of the study. Although polysomnographic recording equipment will be different in each participating center, standard scoring rules will be implemented [53,56,57,58,59]. The morning after testing, parents will be requested to fill out a subjective questionnaire about the quality and quantity of that night’s child sleep. Scoring of respiratory events will be conducted using the same rules in PSG and HRP (see below). 

#### 2.4.2. Home Respiratory Polygraphy (HRP)

Respiratory events will be scored using the same criteria in both HRP and PSG, except for the presence of a respiratory arousal which will only be identified in the PSG. In addition, derivation of AHI will be based on total recording time in HRP and on total sleep time in PSG.

The HRP will be scheduled based on the usual bedtime of the child as obtained from the sleep questionnaire. A recommended time in bed during the HRP of ≥8 h will be requested from participants. A HRP recording will be deemed valid if recording duration is ≥5 h. In both HRP and PSG recordings, sections with artifacts or poor signal will be excluded from analysis. In the event that a HRP or PSG is not valid, it will be repeated within the next 7 d. 

The HRP system to be implemented is the Apnea Link Air (Resmed^®^ Spain, (Madrid, Spain) (Figure 2). This is a type III home sleep apnea testing (HSAT) device that has been validated in the diagnosis of OSA in adults, but not in children [34,35,36]. ApneaLink Air measures being recorded include nasal airflow and snoring using a nasal pressure cannula, blood oxygen saturation, and heart rate by pulse oximetry, and respiratory effort using a thoracic piezoelectric chest belt. The device is small and light (~50 g; 125 × 60 × 30 mm). The software allows to carry out both automated and manual analyses.

Parents will be provided with the device to take home with a prior detailed explanation on how to operate it, along with careful instructions on the correct placement of the device sensors and monitoring of the child. Parents will also perform multiple trials to become comfortable in the device operation guidelines. When the patients return the device the following day, the raw data files will be transferred to a computer and scored automatically and manually by an experienced sleep technologist, while excluding artifact periods.

In an effort to assess the inter-scorer variability between centers, all the recordings for both PSG and HRP will also be scored by the same technician in the study reference center.

### 2.5. Questionnaires

Several questionnaires will be included as follows:(a)Sleepiness rating log [60](b)Pediatrics sleep questionnaire (PSQ) [61](c)“Bears” (Bedtime, Excessive daytime sleepiness, Awakening, Regularity, Sleep-disordered breathing) test [62](d)“Children´s Sleep Habits Questionnaire—CSHQ” and restless legs syndrome questionnaire [63](e)Kiddy-KINDL quality of life of children and young people test (version for parents) [64](f)“Pictorial-Based Declarative Memory Questionnaire” (PDMQ [65]. This test consists to children were then shown a booklet containing a series of 30 colorful pictures of common animals that are highly familiar to children (Figure 3). The child initially identifies the animal being shown and then the investigator also names each animal as further corroboration of the adequate recognition of the animal on each picture, while pointing them out page after page. Subjects were allowed 10 s to look each animal picture. The book was then closed, and the subjects were given 2 min to freely recall any of the animals they could remember without looking at the pictures.

All tests will be completed on two occasions, during the initial and follow-up PSG in the morning after the sleep study. Moreover, the PDMQ will be performed twice, i.e., the night before beginning the sleep study and the morning after each PSG.

### 2.6. Urine Sample Collection, Processing and Analysis 

Morning first void mid-stream urine samples will be collected after each of the two PSG in two 50 mL aliquots that will be immediately stored at −70 degrees until their analysis in the laboratory of the Department of Child Health at the University of Missouri School of Medicine at Columbia, MO, USA. After appropriate aliquoting and initial centrifugation and vortexing, isolation of exosomes will be performed as previously reported [66,67]. Circulating miRNAs are in body fluids such as in urine and are packaged in secreted urinary extracellular vesicles which protect them from degradation. The exosomal fraction will be extracted using ultracentrifugation and size exclusion chromatography harvested from 20 mL urine, and will be subjected to further characterization using western blotting and transmission electron microscopy, and quantification based on nanoparticle tracking analysis using NanoSight NS300 (405 nm laser diode) according to the manufacturer’s protocols (Malvern Instruments Inc.). The Norgen Biotek’s Urine Cell-Free Circulating RNA Purification Maxi Kit (Cat. 56600) will be used to isolate miRNAs from 50 mL urine samples according to manufacturer’s instructions. The mirVana miRNA Isolation Kit (Life technologies, AM1560) will be used for miRNA isolation from the urinary exosomes. Exosomes are lysed in the equal volume of sample/lysis buffer and further processed according to manufacturer’s protocol. RNA concentration and quality will be assessed using Nanodrop 1000 spectrophotometer (Thermo Scientific), and Agilent RNA 6000 Nano Kit, Agilent Small RNA kit on Agilent 2100 Bioanalyzer (Agilent Technologies). cDNA synthesis and qPCR will be performed using the TaqMan Advanced miRNA cDNA Synthesis Kit (Applied Biosystems, A28007) according to manufacturer’s instructions. MicroRNA advanced TaqMan assays will then be used for qPCR and results will be analyzed with web browser-based Applied Biosystems™ Real-Time PCR Analysis Modules.

### 2.7. Ethics and Safety Concerns

This study is not anticipated to result in any meaningful health risks for children, except for the discomfort of performing two consecutive sleep tests within a week of each other. In the HRP test, the children will sleep in their usual bedroom at home and family member returns the equipment the next day. Questionnaires are completed the night of the PSG in the sleep center and urine collection is obtained then in the morning. The second six-month sleep PSG will be performed following the usual clinical treatment protocol for those children with OSA, but reflect an additional PSG for those in whom OSA was not present in the initial study. As indicated above, the project was approved by the ethics committees of each of the participating centers. The parents will sign an informed consent, and for children ages > 7 years, an informed assent will be also signed by the child. Finally, although the study lacks any anticipated risks, all subjects will be covered by the general insurance policy of the Osakidetza-Basque Health Service and by the National Health System.

## 3. Statistical Analysis

All data will be collected in an electronic Case Report Form (eCRD).

Sample size: We estimated a prevalence of 60% of symptomatic children with an AHI > 3 events/h. Thus, a total of 320 eligible subjects are needed assuming a 10% attrition, and a difference in effectiveness between PSG and HSP that will not exceed 9% (α = 0.05 and a β coefficient of 90%). Subjects who refuse to participate and those identified as ineligible will be recorded.

Cost effectiveness analysis is proposed as a non-inferiority study, comparing baseline PSG and 6-month follow-up PSG derived AHI (alpha 0.05; 80% power). It is assumed that the non-inferiority limit is 0.45 units of AHI, i.e., the PSG AHI at 6 months follow-up for therapeutic decisions made with either the PSG or the HPR findings, with a cut-off AHI set at 3.6 ± 2.0 events/h [38]. Considering attrition, 155 subjects will be required to complete the study in each arm (PSG and HPR).

Data will stored in a networked Deductive Database (BBDD) specifically designed and maintained for the study to this study. Statistical analysis will be performed with SPSS package (IBM, New York, NY, USA version 22.0). The measure of the overall agreement between the different categories of the AHI as measured by PSG and HRP, in their automatic and manual analysis, will be assessed by establishing the diagnostic accuracy of the scores obtained by each of the systems. Different categories will be established according to the OSA diagnostic criterion cut-off (AHI > 3 events/h). The HRP discriminatory capacity for the diagnosis of exclusion and confirmation of OSA will be estimated according to different severity AHI cut-off values using ROC curves, area range under the curve and subsequent Bayesian analysis, as previously reported [15,68]. Therapeutic decisions made with PSG will be compared with decisions made with HPR, with data being interpreted independently by all participating centers and compared with each other using Cohen’s Kappa. Further analyses will be carried out on both arms based on intent to treat. In cases considered as attrition, imputation values will be allocated using multiple regression.

Financial analyses will include only direct costs: (a) Costs of testing (HPR and PSG) including the following items: personnel (technical, medical, nurses, and secretaries); amortization of HRP and PSG equipment based on the assumption of 200 studies per year for 5 years; fungible material; and costs across the medical center as a whole; (b) Cost of patients: cost of transfer and return patients to hospital for PSG and HRP; (c) Cost of PSG and HRP test repetitions as needed due to technical failure or unsatisfactory test; (d) Cost of CPAP treatment (Figure A1), adenotonsillar surgery, orthodontics, and medical treatment; and (e) cost of health care after randomization (number of visits in primary care, specialized care and emergency or hospital care excluding those visits related to adenotonsillar surgery; drugs (trade names) and inpatient costs (number of patients and hospital days).

Costs associated with each of the diagnostic approaches will be assessed against the effectiveness of the primary variable (AHI) using Bayesian cost-effectiveness analysis techniques. The decision-making measures studied will consist of the cost-effectiveness ratio of each treatment, the incremental cost-effectiveness ratio (ICER), the cost-effectiveness plane, the net profit of each treatment, the incremental net profit (NB) and the cost-effectiveness acceptability. We will apply sensitivity analysis using decision model simulation methods. Specifically, we have developed a probabilistic sensitivity analysis of the cost-effectiveness ratio by specifying uncertain parameter distributions in our decision model. To estimate probability intervals, we categorize numerical values of ICER in order to consider preferences on the cost-effectiveness plane.

## 4. Study Significance

Currently, the gold standard approach to the diagnosis of OSA in children remains a PSG in the laboratory [12]. However, the emergence of a large array of home-based technologies and the successful implementation of HST in adults has prompted substantial interest in expanding the role of HST to the pediatric population. Such need is all the more pressing considering on the one hand the elevated prevalence of children who habitually snore and are therefore at-risk for OSA [69,70,71,72,73], and on the other hand the paucity of pediatric sleep laboratories, the scarcity of pediatric sleep medicine physicians, the inconvenience for the child and their family, the high costs in labor, and of the facilities underlying PSG testing. However, HST approaches are not yet extensively validated in children, and their usefulness and financial advantages have not been thoroughly explored despite the relatively large number of published studies comparing PSG and HST.

The proposed study incorporates not only the validation in a real life setting of a HST approach as a diagnostic tool, but also includes an assessment of its robustness in the clinical decision making of when to provide clinical treatment and when to abstain from a therapeutic intervention. Furthermore, it will examine not only the comparative outcomes obtained with HST and PSG, but will also evaluate cognitive outcomes as illustrated by a declarative memory test [65]. In addition, it will provide estimates of feasibility of automatic scoring that should alleviate the laborious burden of manual scoring of HST in children. Importantly, economic assessments of the HST relative to the PSG will be conducted.

Finally, a simple and non-invasive method such as the proposed examination of exosomal miRNAs as biomarkers of OSA in children, and also as biomarkers of cognitive deficits may further advance the field of biomarkers in sleep medicine [46], while enabling initial steps for a convenient diagnostic tool in children with suspected OSA that also carries enabling elements of precision medicine [74,75,76].

## Figures and Tables

**Figure 1 mps-04-00009-f001:**
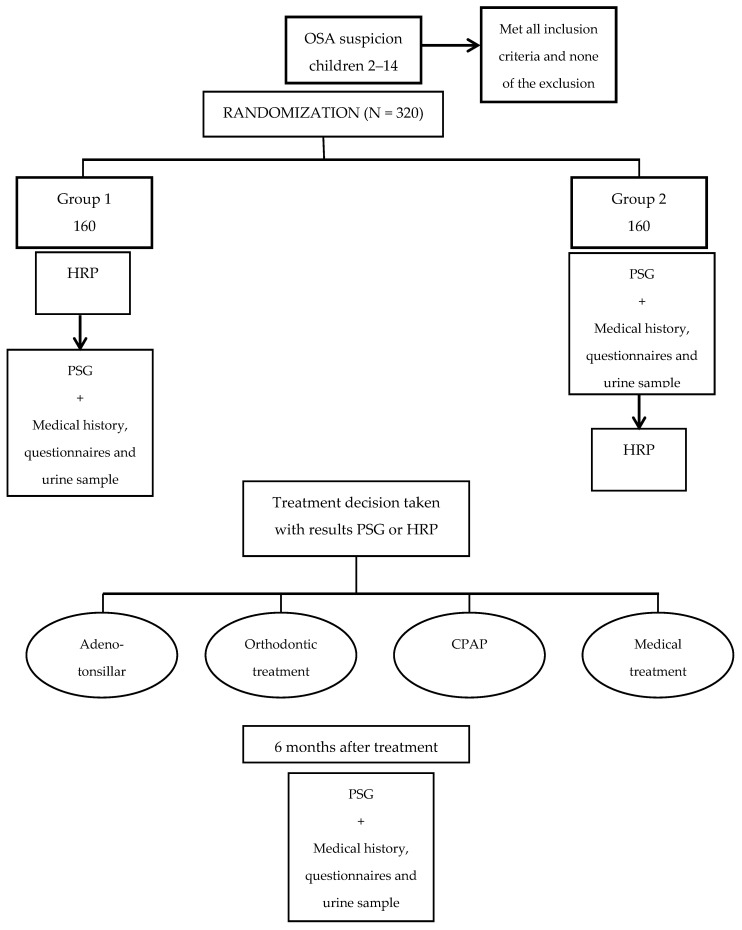
Flow chart and follow up of randomization of children.

**Figure 2 mps-04-00009-f002:**
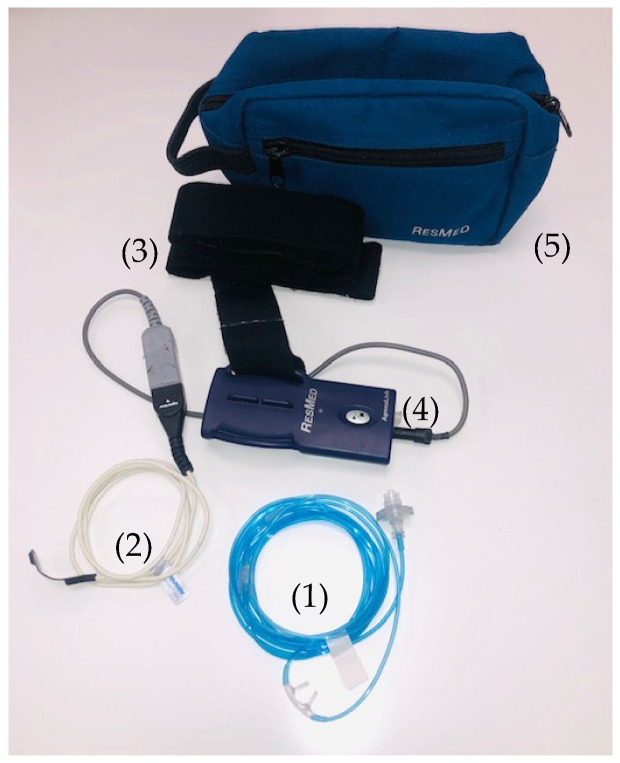
Apnea Link Air polygraphic system (Resmed^®^Spain): (1) Pressure nasal canula; (2) Pulse oximeter; (3) Piezoelectric chest belt; (4) Registration box; (5) Transport bag. (See methodology section).

**Figure 3 mps-04-00009-f003:**
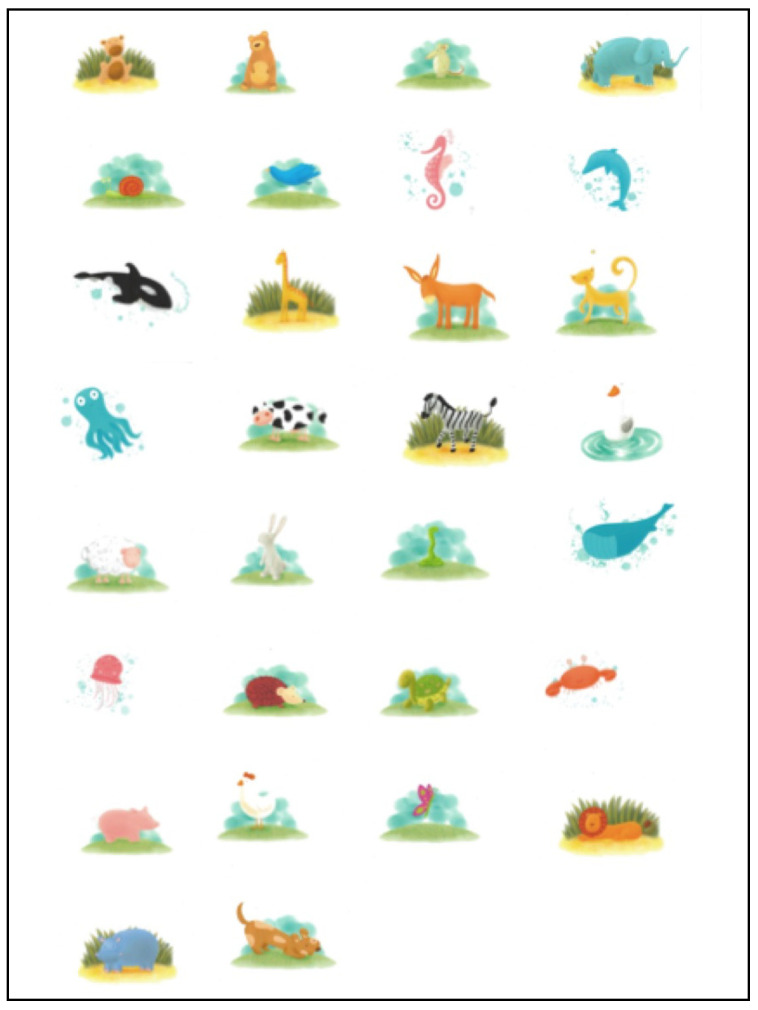
Pictorial-Based Declarative Memory Questionnaire [61]: Series of 30 colorful pictures of common animals that children should remember without looking at the pictures (see methodology section for more details).

**Table 1 mps-04-00009-t001:** Inclusion/Exclusion criteria.

Inclusion Criteria	Exclusion Criteria
Children ages 2 to 14 years, both sexes	Inability to perform the HST
Children with clinical suspicion of OSA (defined in text)	Severe or unstable underlying cardiovascular, renal, or respiratory disease
Children referred for a sleep study for suspected OSA	Children with genetic craniofacial syndromes, Down syndrome, or neuromuscular diseases
Informed consent (parents or legal guardian children)	Children with chronic insomnia and/or depression, autism, psychiatric condition, developmental delay
	Complete or almost complete nasal obstruction that prevents obtaining a quality signal with the HRP
	History of surgery for OSA and/or previous treatment with continuous positive airway pressure.
	Place of residence more than 100 km from the sleep center
	Abbreviations: OSA, obstructive sleep apnea, HRP: home respiratory polygraphy

**Table 2 mps-04-00009-t002:** Itemized list of variables to be collected.

Type of Variable	Description
Clinical and anthropometric variables	Age, gender, and comorbidities. Physical examination: height, weight and corresponding percentiles, body mass index (BMI) and z score, neck and waist circumference, arterial blood pressure (ABP), degree of tonsil hypertrophy, Mallampati score, and maxillomandibular profile and type of bite.
HRP and PSG parameters	Sleep parameters (only PSG) Respiratory events Oximetry measures CO_2_ parameters (only PSG) Snore Body position Heart rate
Questionnaires	Sleepiness rating log Pediatric sleep questionnaire (PSQ) Bears test (Bedtime, Excessive daytime sleepiness, Awakening, Regularity, Sleep-disordered breathing) CSHQ (Children’s Sleep Habits Questionnaire) Kiddy-KINDL quality of life of children. Pictorial-Based Declarative Memory Questionnaire
Morning First void urine sample	50 mL aliquots preserved at −80 °C.

**Table 3 mps-04-00009-t003:** 2015 AASM Guidelines for the scoring of pediatric respiratory events [1].

Respiratory Event	Definition
Obstructive apnea	Drop in airflow by 90% of pre event baseline for ≥2 breaths associated with respiratory effort
Central apnea	Drop in airflow by 90% of pre-event baseline with absent inspiratory effort and event lasts ≥20 s or ≥2 breaths and associated with arousal or 3% oxygen desaturation.
Mixed apnea	Apnea for >2 breaths with periods of both absent and active respiratory effort, in either sequence.
Hypopnea	Drop in peak flow by ≥30% with either ≥3% oxygen desaturation or arousal for >2 breaths.
RERA	At least 2 breaths when the breathing sequence is characterized by increasing respiratory effort, flattening of the inspiratory portion of the nasal pressure flow waveform, snoring, or an elevation in the end-tidal PCO_2_ leading to arousal from sleep when the sequence of breaths does not meet criteria for an apnea or hypopnea.
Hypoventilation	CO_2_ > 50 mm Hg for >25% of total sleep time, measured by end-tidal CO_2_ or CO_2_ transcutaneous.

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
