# Peer review of "Validity and Cost-Effectiveness of Pediatric Home Respiratory Polygraphy for the Diagnosis of Obstructive Sleep Apnea in Children: Rationale, Study Design, and Methodology"

_mps, 2021, doi:10.3390/mps4010009_

Round 1

Reviewer 1 Report

Manuscript ID: mps-1074983

Title: "VALIDITY AND COST-EFFECTIVENESS OF PEDIATRIC HOME RESPIRATORY POLIGRAPHY FOR THE DIAGNOSIS OF OBSTRUCTIVE SLEEP APNEA IN CHILDREN: RATIONALE, STUDY DESIGN, AND METHODOLOGY"

The aim of this protocol was to validate the use of Home Respiratory Polygraphy (HRP) in paediatric population to assess Obstructive Sleep Apnoea (OSA) in children.

The manuscript is really interesting and well written.

Methodology is consistent with conventionally recognized standards.

I only have few suggestions and/or naïve questions.

At page 3, there is probably an “intruder” paragraph from line 107 to line 116.

Authors suggested to administer several questionnaires, however, the rationale for so many instruments is not presented. Since the time necessary to fill in them is not secondary, I think it could be good to motivate their use.

At page 10 (line 353), Authors wrote “for children ages >7 years, an informed assent will be signed by the child”. It is not clear to me whether in this case parents are excluded by this procedure. Probably this sentence could be replaced by “for children ages >7 years, an informed assent will be also signed by the child”.

At page 11 (line 368), Authors wrote “a cut-off AHI set at 3.6±2.0”. In my opinion a cut-off value should be a specific value, not a range.

In the last section (page 11 and 12) the acronym HST is used, while I think it should be HPR.

Reviewer 2 Report

I have read the article by Oceja et al. with great interest. I agree that there is an unmet need for home diagnostic sleep tests in paediatric OSA.

Comments:

  • Introduction and Methods. I think ApneaLink also records snore tracing which could really be helpful to determine when the patient was asleep. I also believe that the device has an accelerometer to determine body position. Please, clarify.
  • Please delete the part between lines 107-116.
  • Table 1. “Both sexes” I would change to “any sex”.
  • Line 250. Please, change “were” to “will be”.
  • Urine biomarkers. I understand the value of measuring them to better understand the pathomechnanism of OSA. but I do not think the authors really intend to use them as a diagnostic tool. They are less feasible and definitely not cost-effective compared to PSG. What is the intent with these data? Are you going to compare them between OSA and controls? Do you want to correlate them with questionnaires, disease severity? Do you want to compare the discrimination utility to home respiratory sleep study? Please, clarify!
